# Biochemical Properties of a Promising Milk-Clotting Enzyme, Moose (*Alces alces*) Recombinant Chymosin

**DOI:** 10.3390/foods12203772

**Published:** 2023-10-13

**Authors:** Dina V. Balabova, Ekaterina A. Belash, Svetlana V. Belenkaya, Dmitry N. Shcherbakov, Alexander N. Belov, Anatoly D. Koval, Anna V. Mironova, Alexander A. Bondar, Ekaterina A. Volosnikova, Sergey G. Arkhipov, Olga O. Sokolova, Varvara Y. Chirkova, Vadim V. Elchaninov

**Affiliations:** 1Institute of Biology and Biotechnology, Altai State University, 656049 Barnaul, Russia; 2State Research Center for Virology and Biotechnology “Vector”, Rospotrebnadzor, 630559 Koltsovo, Russia; 3Federal Altai Scientific Center for Agrobiotechnologies, Siberian Research Institute of Cheese Making, 656910 Barnaul, Russia; 4JCF “Genomics”, Institute of Chemical Biology and Fundamental Medicine, Siberian Branch of the Russian Academy of Sciences, 630090 Novosibirsk, Russia; 5Boreskov Institute of Catalysis, Siberan Branch of the Russian Academy of Sciences, 630090 Novosibirsk, Russia

**Keywords:** recombinant chymosin, moose, milk-clotting activity, thermal stability, proteolytic activity, coagulation specificity, Michaelis–Menten kinetics, temperature optimum, calcium chloride concentration, substrate pH

## Abstract

Moose (*Alces alces*) recombinant chymosin with a milk-clotting activity of 86 AU/mL was synthesized in the *Kluyveromyces lactis* expression system. After precipitation with ammonium sulfate and chromatographic purification, a sample of genetically engineered moose chymosin with a specific milk-clotting activity of 15,768 AU/mg was obtained, which was used for extensive biochemical characterization of the enzyme. The threshold of the thermal stability of moose chymosin was 55 °C; its complete inactivation occurred after heating at 60 °C. The total proteolytic activity of moose chymosin was 0.332 A_280_ units. The ratio of milk-clotting and total proteolytic activities of the enzyme was 0.8. The K_m_, k_cat_ and k_cat_/K_m_ values of moose chymosin were 4.7 μM, 98.7 s^−1^, and 21.1 μM^−1^ s^−1^, respectively. The pattern of change in the coagulation activity as a function of pH and Ca^2+^ concentration was consistent with the requirements for milk coagulants for cheese making. The optimum temperature of the enzyme was 50–55 °C. The introduction of Mg^2+^, Zn^2+^, Co^2+^, Ba^2+^, Fe^2+^, Mn^2+^, Ca^2+^, and Cu^2+^ into milk activated the coagulation ability of moose chymosin, while Ni ions on the contrary inhibited its activity. Using previously published data, we compared the biochemical properties of recombinant moose chymosin produced in bacterial (*Escherichia coli*) and yeast (*K. lactis*) producers.

## 1. Introduction

An aspartic milk-clotting endopeptidase—chymosin (EC 3.4.23.4)—is synthesized in the stomach of newborns of most species belonging to the class Mammalia [1]. Among gastric pepsin-like proteinases, chymosin (Chn) is distinguished by an unusual combination of enzymatic properties; it exhibits high specificity with respect to a single peptide bond in the kappa-casein (κ-CN) molecule and low total proteolytic activity (PA), which determines its biological functions. Firstly, Chn effectively coagulates milk, which contributes to its full assimilation in the gastrointestinal tract of newborns [2,3]. Secondly, the low total PA of chymosin prevents damage to antibodies and other proteins contained in mother’s milk that have antibacterial and antiviral properties. As a result, newborns of those mammals that have not formed their own immune system by the postnatal period are provided with maternal factors of passive immunity. In species whose young are born with already developed immunity, the Chn gene is not expressed [4].

Due to its unique enzymatic and biochemical properties, Chn is widely demanded in cheese making. Chymosin is the main component of rennet—a complex set of milk-clotting enzymes obtained from the stomachs of young ruminants. The use of rennet ensures the maximum yield and highest quality of produced cheeses. However, in the second half of the XIX century, the production volumes of rennet were not enough to meet the needs of the cheese-making industry, and its genetically engineered analogues replaced the natural Chn [5].

The production of cow (*Bos taurus*) Chn in genetically modified microorganisms and its implementation in cheese making was one of the first successful applications of recombinant DNA technologies in the food industry [6,7,8]. Later, there appeared recombinant Chn (rChn) of one-humped camel (*Camelus dromedarius*), which is superior to the cow enzyme in terms of specific milk-clotting activity (MA) and specificity (MA/PA) [9,10]. Genetically engineered cow and camel Chns are a high-quality alternative to rennet and currently are widely represented in the market of industrial milk coagulants [11].

Further improvement of cheese-making technologies requires the search for new, more effective milk-clotting enzymes (MCEs). Therefore, one of the urgent tasks of modern biotechnology is to obtain milk-clotting proteinases that exceed cow and camel rChns in technological properties. One of the ways to solve the problem is to study Chns of various mammals [12].

From 2001 to the present, the following rChns have been obtained and characterized: sheep (*Ovis aries*) [13], goat (*Capra hircus*) [14,15,16], buffalo (*Bubalus arnee bubalis*) [17,18], yak (*Bos grunniens*) [19,20], alpaca (*Vicugna pacos*) [21], Altai maral (*Cervus elaphus sibiricus*) [22,23,24,25], rabbit (*Oryctolagus cuniculus*) [26], white whale (*Delphinapterus leucas*) [27], two-humped camel (*Camelus bactrianus*) [28,29], and tupaya (*Tupaia belangeri chinensis*) [30].

Earlier, moose (*Alces alces*) rChn was obtained in the *Escherichia coli* expression system (SHuffle Express strain), which demonstrated some biochemical properties attractive from the point of view of cheese production [31]. Unfortunately, this expression system has some drawbacks: the rChn zymogen synthesized in *E. coli* accumulates intracellularly in the form of insoluble inclusion bodies, which hinders obtaining an active enzyme. Besides, prokaryotic producers cannot provide post-translational processing of synthesized proteins.

To obtain a post-translationally modified enzyme and further study its biochemical properties, we set the task of synthesizing moose rChn in the eukaryotic expression system.

Chymosin genes can be expressed by various eukaryotic producers: plants [32,33], fungi [34], and mammalian cells [35].

The expression systems of unicellular fungi (yeast) are characterized by the simplicity of genetic manipulation and stable genomic integration of expression cassettes, the ability to carry out posttranslational modifications (PTMs), the secretion of target proteins, and the absence of viral infections, toxins, pyrogens, and pathogens [36,37,38]. Among other yeasts, *Kluyveromyces lactis* exhibits the following attractive properties: the Crabtree-negative carbohydrate metabolism, the ease of scaling, the GRAS status, the possibility of cultivation using simple culture media, and the availability of highly efficient promoters. All these features make yeast *Kluyveromyces lactis* a convenient tool for obtaining heterologous recombinant proteins both on a laboratory and industrial scale [37,39,40]. To obtain the enzyme, in our work, we designed an original expression cassette that included a previously developed hybrid promoter [40]. For the first time, this demonstrated the possibility of using this promoter for the synthesis and secretion of moose prochymosin in *K. lactis*.

The purpose of this work was to obtain a detailed biochemical characterization of moose rChn obtained in the *K. lactis* production system (rChn-Alc-KL).

## 2. Materials and Methods

### 2.1. Construction of an Integrative Plasmid Vector

To express the moose prochymosin gene (ProChn-Alc) in *K. lactis* cells, we constructed an integrative vector containing the zeocin resistance gene as a selectable marker. The ProChn-Alc nucleotide sequence (GenBank MT542132), synthesized by “DNK Sintez” (Russia), was inserted into the integrative vector pSVB at the unique restriction sites of BamHI and AspA2I. As a result, the obtained pSVB-Alc vector (Figure 1) made it possible to integrate an expression cassette containing the ProChn-Alc gene into the genome of a commercial *K. lactis* strain GG799.

### 2.2. Preparation of Kluyveromyces Lactis Strain Alc-D Transgenic for Moose ProChn

Plasmid pSVB-Alc was treated with the ZrmI enzyme (“SibEnzyme”, Novosibirsk, Russia), and *K. lactis* cells GG799 were transformed with 1–5 μg of DNA using a Gene Pulser Xcell TM electroporator (“Bio-Rad”, Hercules, CA, USA) at 1500 V, 25 μF, 400 Ω, and a pulse duration of 0.01 s. The transformants were selected on the YCB medium, which included agar (20.0 g/L), the antibiotic Zeocin (“InvivoGen”, San Diego, CA, USA) at the concentration of 750 µg/mL, and two components from the *K. lactis* Protein Expression Kit (“NEB”, Ipswick, MA, USA)—YCB Medium powder (11.7 g/L) and Acetamide solution (10.0 mL/L). The presence of the ProChn-Alc gene in selected colonies was confirmed by PCR. As a result, a producer of recombinant moose prochymosin (rProChn-Alc)—*K. lactis* Alc-D—was obtained.

### 2.3. Producer Cultivation

The producer strain was grown in the YEP culture medium (composition, g/L: yeast extract, 10; peptone, 20) and its variants with different glucose contents (10, 20, and 30 g/L). In addition, the culture medium was supplied with calcium pantothenate (1.6 mg/L) and KH_2_PO_4_ (1 g/L).

Colonies of *K. lactis* Alc-D cells were added to 5 mL of the YEP medium and left in a shaker-incubator at 30 °C and 200 rpm for 24 h. The resulting inoculum, in a ratio of 1/100, was transferred to an Erlenmeyer flask containing 150 mL of the YEP medium and cultivated at 30 °C and 250 rpm for 3 days. Glucose concentration in the culture medium was monitored using a Diacont glucometer (“OK Biotech”, Taiwan). After glucose depletion, cultivation was continued for 9 or 15 days. Upon cultivation, producer cells were sedimented by centrifugation (20 min, 5000× *g*, 4 °C) to obtain a culture fluid (c.f.) containing rProChn-Alc.

The conditions for obtaining the target protein were selected according to the same algorithm, by varying the duration of cultivation and the concentration of glucose.

### 2.4. Zymogen Activation

rProChn-Alc was activated according to the procedure described in [23]. As a result of activation, *K. lactis* c. f. containing rChn-Alc-KL was obtained.

### 2.5. Partial Purification of Moose rChn

(NH_4_)_2_SO_4_ (40 g per 100 mL) was added to the c.f. containing active rChn-Alc-KL, stirred for 30 min, and left at 4 °C for 20 h. Then the precipitate was separated by centrifugation (13,130× *g*, 4 °C, 40 min), dissolved in buffer A (50 mM Na-acetate buffer, pH 5.0, 2.0 M NaCl), and applied to a Phenyl Seplife HP column (“Sunresin”, Shaanxi, China). The column was washed with the buffer until the release of proteins that did not bind to the hydrophobic sorbent. Fractions containing rChn-Alc-KL were eluted with buffer B (50 mM Na-acetate buffer, pH 5.0), combined, and applied to a Heparin Sepharose HP column (“GE Healthcare”, Salt Lake, UT, USA). The column was washed with buffer B until the removal of the material that did not bind to the carrier. The target enzyme was eluted with buffer B containing 0.25 M NaCl.

Total milk-clotting activity, total proteolytic activity, thermal stability, and dependence of the coagulation time on CaCl_2_ concentration and pH were determined according to [21].

### 2.6. Biochemical Characterization of rChn

#### 2.6.1. Enzymatic Kinetics

Michaelis–Menten kinetics were studied using fluorescence spectroscopy. The following parameters were determined: the Michaelis constant (K_m_), the maximum rate of the enzymatic reaction (Vmax), the turnover number of the enzyme (k_cat_), and the catalytic efficiency (k_cat_/K_m_). The measurements were carried out using a CLARIOstar microplate reader (“BMG LABTECH”, Ortenberg, Germany) according to a procedure described in [23].

#### 2.6.2. Dependence of MA on the Temperature of the Substrate (Temperature Optimum)

The substrate was whole pasteurized milk, into which NaN_3_ was added to the final concentration of 0.02%. The pH was adjusted to 6.5, and the substrate (1.25 mL) was heated for 10 min in a water bath at a specific temperature in the range of 25–65 °C with an interval of 5 °C. After adding 0.1 mL of rChn, the coagulation time (CT) was determined in seconds. The MA of the enzymes was determined as MA = 1/CT and expressed in %. The maximum MA was taken as 100%, and the dependence of MA (%) on the temperature of the substrate was plotted.

#### 2.6.3. Influence of Divalent Metal Cations on Milk-Clotting Activity

In these experiments, the substrate was whole unpasteurized milk. NaN_3_ was added to it to set the final concentration of 0.01%, and the pH was adjusted to 6.5. Aliquots of the substrate (50 mL) were mixed with a 1.0 M solution of CuCl_2_, MgCl_2_, NiCl_2_, ZnCl_2_, CoCl_2_, BaCl_2_, FeCl_2_, MnCl_2_, or CaCl_2_ to the final concentration of 10 mM, and CT was determined after the addition of the studied rChn. The milk-clotting activity of the enzymes was also calculated as MA = 1/CT and expressed in %. The MA of the substrate with no salt added was taken as 100%.

#### 2.6.4. Determination of Proteolytic Specificity by Electrophoresis

The substrate was whole unpasteurized cow’s milk. It was diluted in a ratio of 1:4 with a 20 mM Na-acetate buffer (pH = 5.65) containing 0.01% NaN_3_. The substrate was used on the day of preparation. A total of 250 μL of the substrate was mixed with 5 μL of rChn with an activity of ≈1000 AU/mL. The resulting enzyme–substrate mixtures were thoroughly stirred and incubated at 35 °C for 1 h. After the incubation, the mixtures were blended with a sample buffer for SDS-PAGE (“Serva”, Heidelberg, Germany) in the ratio of 1:1 and heated in a boiling-water bath for 90 s. Then, the samples were studied using electrophoresis in the presence of sodium dodecyl sulfate (SDS-PAGE) [41]. The LMW-SDS Marker Kit (“GE Healthcare”, USA) was used as molecular mass (MM) markers. In the control samples, instead of a rChn solution, 5 μL of a 20 mM Na-acetate buffer (pH 5.65) containing 0.01% NaN_3_ was added. The control samples were incubated for 1 h either at room temperature (control 1) or at 35 °C (control 2).

#### 2.6.5. Preparations of Cow and Camel rChns

A concentration of 0.5% aqueous solutions of commercial cow rChn (rChn-Bos) and one-humped camel rChn (rChn-Cam) produced by “Chr. Hansen” (Hørsholm, Denmark) were used as reference samples. The biochemical properties of the enzymes were determined using rChn-Alc-KL, rChn-Bos, and rChn-Cam samples normalized by their MA. All measurements were repeated at least three times (*n* ≥ 3).

#### 2.6.6. Protein Concentration

Protein concentration in rChn preparations was determined spectrophotometrically according to Warburg and Christian [42].

### 2.7. Statistical Processing

The statistical processing of the experimental data was conducted with the use of Microsoft Excel. Quantitative variables were presented as the arithmetic mean (M) with standard deviation (±SD). The graphs did not indicate a 95% confidence interval if its values were less than 10% of the variable value.

### 2.8. Crystallization of Moose rChn and Acquisition of X-ray Diffraction Data

Moose rChn was crystallized by the sitting drop method using an NT8—Drop Setter/Crystallization Robot (“Formulatrix”, Bedford, MA, USA). With the use of the PACT premier solutions (“Molecular Dimensions”, London, UK) [43], the following crystallization conditions were selected: 0.2 M CaCl_2_, 0.1 M sodium acetate (pH 5.0), 20% (*w*/*v*) PEG 6000. The diffraction data from the obtained single crystals were collected at the ID23-1 beamline of the European Synchrotron Radiation Facility (Grenoble, France) within the MX2270 project. Primary diffraction data were obtained in the φ-scan mode with a step of 0.05° and processed in the XDS software (version 5 February 2021) [44].

## 3. Results and Discussion

### 3.1. Construction of the Producer Strain

The yeast producer of moose rProChn was obtained using the *K. lactis* strain GG799 with an integrative plasmid vector pSVB-Alc (Figure 1). Before cloning in the pSVB vector, the codon composition of the moose prochymosin gene was optimized for expression in *K. lactis*. Controlled expression of the target gene in the recombinant producer was achieved using a hybrid autoinducible P_350_ promoter, which included the regulatory sequence of the isocitrate lyase promoter and the core sequence of the glyceraldehyde 3 phosphate dehydrogenase promoter [24,40]. An important feature of the P_350_ promoter is its activation upon glucose depletion in the culture medium.

The zeocin resistance gene served as a marker for the selection of transformants. The producer strain Alc-D *K. lactis* was obtained as follows: a suspension of *K. lactis* cells GG799 was electroporated in the presence of the recombinant pSVB-Alc plasmid and then inoculated on YCB selective agar medium, containing acetamide as the only source of nitrogen.

### 3.2. Cultivation of the Producer Strain

The initial glucose content in the culture medium significantly impacts the yeast culture’s density and its productivity in terms of the target protein. Glucose is rapidly consumed by growing cells, and after its depletion, the P_350_ promoter provides the synthesis of rProChn-Alc, which is secreted into the culture medium [40].

There were two sets of experiments. In the first set, the effect of glucose concentration on the synthesis of the target protein was studied. For this, the culture medium was supplied with three different concentrations of glucose: 1% (series 1), 2% (series 2), and 3% (series 3). To evaluate the efficiency of rProChn-Alc production, during the cultivation of the producer, we took aliquots of the c.f., activated the zymogen in them, and determined the MA of the preparations obtained.

After 3 days of cultivation, rChn-Alc-KL in the c.f. was not detected regardless of the concentration of glucose. Cultivation for 6 days led to the appearance of MA in the c.f. with all concentrations of glucose. At that, the highest yield of the target protein (77.4 ± 0.7 AU/mL) was obtained at the initial glucose concentration of 2%. After 9 days of cultivation, preparations of series 1 and 2 showed a drop in the total MA by 1.6 and 2.9 times, respectively. On the contrary, the producer with the initial glucose concentration of 3% continued to increase its MA (Figure 2a).

In the second set of experiments, moose rChn was cultivated in a medium containing 3% glucose, 1 g/L KH_2_PO_4_, and 1.6 mg/L calcium pantothenate. Under these conditions, the highest MA (85.5 ± 7.0 AU/mL) was recorded on the 15th day of cultivation (after 15 days, the activity of the target protein began to decrease and cultivation was stopped). A typical dependence of the MA of the c.f. on the duration of cultivation is shown in Figure 2b.

### 3.3. Activation of the Zymogen and Yield of the Target Enzyme

Immediately after the deposition of the Alc-D *K. lactis* biomass, the coagulation activity of the c.f. was less than 0.5 AU/mL. As a result of activation, the MA of the preparation increased by more than two orders of magnitude and achieved ≈86 AU/mL, which indicated the conversion of rProChn into active moose rChn.

### 3.4. Partial Purification of Moose rChn

As a result of partial purification using salting out, hydrophobic, and affinity chromatography, the relative MA (MA/A_280_) of the target enzyme increased by approximately 2450 times (Table 1).

### 3.5. Milk-Clotting Activity

To determine the main biochemical properties, the MA of moose rChn preparations and reference enzymes was adjusted to 1200–1400 AU/mL.

As seen from Table 2, the specific MA of rChn-Alc-KL occupied an intermediate position between the MAs of rChn-Bos and rChn-Cam.

According to [31], the coagulation activity of moose rChn obtained in the *E. coli* expression system (rChn-Alc-EC) was 37% of that of cow rChn. Thus, the replacement of the prokaryotic expression system with the eukaryotic one resulted in a 2.9-fold increase in the specific MA of moose rChn.

### 3.6. Thermal Stability

The ability to resist denaturation at increased temperatures is the most important biochemical characteristic of any enzyme.

However, unlike that of most technological enzymes, the thermal resistance of milk coagulants used in cheese making should not tend to the maximum. An exemplary MCE for cheese production should exhibit a high MA at 32–33 °C, providing the formation of a curd, and undergo maximum inactivation above 40 °C, during the heat treatment of cheese grain. The MCE inactivation after milk clotting improves the technological performance of cheese whey, which is used as a raw material for food production and reduces the intensity of nonspecific proteolysis in cheeses. Among commercial genetically engineered enzymes, the thermal stability closest to optimal is demonstrated by cow rChn [45].

Thermal stability (TS) of a milk coagulant should be examined in relation to its total PA, which is considered as a negative factor in cheese-making technologies [46]. According to [9], with an increase in the temperature of enzyme–substrate mixtures, rChns exhibit an increased total PA. This means that heat-resistant MCEs remaining in the curd and having excessive PA can adversely affect the yield and quality of cheeses [47,48].

When cheese is made using cow rChn, an increase in the curd temperature from 50 °C to 56 °C leads to a significant decrease in the concentration of one of the main proteolysis products of αs1-casein (αs1-CN)—water-insoluble (hydrophobic) polypeptide αs1-I (f24-199) [49].

The thermal stability of camel rChn is somewhat higher than that of cow rChn; the melting points (T_m_) of these enzymes are 60.7 and 57.7 °C, respectively [10]. However, even such a small difference in T_m_ (3.0 °C) affects their technological properties. At a second heating temperature of 56 °C in cheeses produced using camel rChn, the intensity of proteolysis was higher than when using cow rChn. Moreover, camel rChn, which has a fourfold lower total PA [9], but exceeds the cow enzyme in TS, causes more intense proteolysis of αs1-CN in ripening and stored cheeses [49].

The literature data indicate that rChn of the same species, but obtained in different producers, can differ in TS [45]. For example, the temperature of complete inactivation of camel rChn synthesized by the mold *Aspergillus niger* [9] and the yeast *Pichia pastoris* [50] differs by 10 °C. The thermal resistance of rChn produced in different strains of the same producer can also vary. For example, buffalo rChn synthesized by *P. pastoris* strain GS115 is completely inactivated at 60 °C [15], while rChn of the same species but produced by *P. pastoris* strain X-33 demonstrates 37% of its initial MA at this temperature [18]. The TS thresholds of maral rChn synthesized in *E. coli* and *K. lactis* were 55 and 40 °C, respectively. In the case of *C. elaphus*, the replacement of a prokaryotic production system with a eukaryotic one led to the synthesis of a highly thermolabile rChn that exhibited equally good performance in the production of cheeses with high and low temperatures of the second heating [51,52].

When investigating the TS of moose and commercial rChns, the thermal inactivation threshold was defined as a temperature at which the enzyme lost more than 20% of its initial MA.

As seen from Figure 3, in the temperature range of 30–45 °C, the residual MA of all enzymes was approximately the same and amounted to 95–100%. At incubation temperatures above 45 °C, rChn-Bos and rChn-Alc-KL began to deactivate, while rChn-Cam retained its properties. After heating at 50 °C, the residual MAs of rChn-Bos, rChn-Alc-KL, and rChn-Cam were 85.5 ± 1.1%, 92.5 ± 1.0%, and 98.7 ± 1.3%, respectively. Even though the thermal inactivation thresholds of cow and moose rChns coincided and amounted to 55 °C, with a further increase in the incubation temperature, the cow enzyme lost its coagulation ability twice as fast. The incubation at 60 °C caused rChn-Alc-KL and rChn-Bos to completely lose their ability to coagulate milk. In contrast, the thermal inactivation threshold of rChn-Cam was 60 °C (residual MA was 63.4 ± 0.9%), and its complete inactivation was observed at 65 °C.

A comparison of the biochemical properties of rChn-Alc-EC [31] and rChn-Alc-KL (this study) shows that the replacement of the bacterial producer by the yeast one does not significantly affect the TS of the enzyme. Indeed, even though rChn-Alc-EC retained approximately 6.5% of its initial MA even after heating at 60 °C [31], the thresholds of thermal inactivation of these enzymes were the same and amounted to 55 °C.

### 3.7. Proteolytic Activity

In the dairy industry, PA of milk coagulants is distinguished into specific and nonspecific. The specific PA is synonymous with MA and aims to hydrolyze the only bond in the molecule of κ-CN (in the case of bovine κ-CN, this is the Phe105-Met106 bond), which causes destabilization and aggregation of casein micelles. The nonspecific (or total) PA is the ability of Chn to hydrolyze any bonds in caseins [10]. The quality of an MCE is evaluated by the ratio of its MA and nonspecific PA (MA/PA), which indicates the coagulation specificity of the enzyme. The higher the MA/PA, the higher is the quality of the MCE, i.e., a perfect MCE should exhibit a maximum MA along with a minimum total PA. In practice, the ratio of MA/PA of bovine CN is considered to be the standard for coagulation specificity [2,9,10].

Since the main function of an MCE in cheese making is the hydrolysis of a single peptide bond of κ-CN [53], it is desirable that the enzyme should not exhibit any PA after milk coagulation. Theoretically, this is possible if the MCE is completely inactivated at the technological stages associated with the heat treatment of the curd. In practice, however, even after the heat treatment at 56 °C, the best commercial rChns do not denature 100% and retain some nonspecific PA, which contributes to the proteolytic degradation of caseins [49,54,55,56]. Moreover, the thermally inactivated MCE remaining in the cheese mass can partially renature and exhibit PA during cheese maturation and storage [57]. In milk and cheese, the products of primary proteolysis of caseins are formed under the action of an MCE and a thermostable serine proteinase—plasmin (EC 3.4.21.7) [56,58,59]. The degradation of caseins to low molecular weight products, which contribute to the organoleptic characteristics of cheeses, is carried out mainly by proteases of the starter microflora [48,60].

At all stages of incubation of enzyme–substrate mixtures, rChn-Alc-KL showed a higher PA than rChn-Bos and rChn-Cam (Figure 4). After 180 min of incubation, the PA of rChn-Alc-KL was 0.332 A_280_ units, which was 1.4 and 9.0 times as high as the PA of rChn-Bos and rChn-Cam, respectively (Table 3).

The coagulation specificities of moose rChn and commercial rChns were compared using the following methodology [9]: specific MA and total PA of the enzymes were expressed as a percentage of the corresponding parameters of cow rChn. The highest coagulation specificity with respect to cow’s milk was demonstrated by camel rChn, and the lowest was demonstrated by moose rChn (Table 3). The specific MAs of moose and cow rChns were similar and differed by only 6%. However, the higher total PA of moose rChn deteriorated its coagulation specificity; the MA/PA ratio of moose rChn was 1.3 and 9.6 times as low as the rChn of cow and camel, respectively.

The literature data on the coagulation specificity of cow and camel rChns are contradictory. On one hand, it was shown that the specificity of camel rChn is about 11% lower than that of cow rChn [15]. On the other hand, our data confirm the results published in [9], according to which rChn-Cam exceeds rChn-Bos by 7.0 times in terms of the MA/PA ratio.

The MA/PA ratio for moose rChn obtained in the *E. coli* expression system was lower than that for the enzyme synthesized in *K. lactis* and was 0.6 [31]. Thus, the replacement of the bacterial producer *A. alces* with the yeast one leads to an increase in the coagulation specificity of moose rChn by 33%.

### 3.8. Proteolytic Specificity

The specificity of rChn with respect to cow’s milk proteins was studied by SDS-PAGE. It was found that the highest proteolytic specificity was demonstrated by camel rChn. After incubation with this enzyme, the electrophoretic profile of milk proteins underwent minimal changes; only κ-CN disappeared and only one of its hydrolysis products (para-κ-CN with MM ≈ 16 kDa) appeared (Figure 5). rChn-Bos and rChn-Alc-KL had similar proteolytic specificity, but in contrast to rChn-Cam, they hydrolyzed not only κ-CN but also some other milk proteins. As a result, in the electrophoretic profiles of preparations obtained with rChn-Alc-KL and rChn-Bos, there appeared additional weak bands with MM = 28–29 kDa and low-molecular-mass polypeptide components with MM << 14 kDa which migrated in the zone of the leading dye. At that, due to the higher total PA, the preparation with rChn-Alc-KL accumulated the highest amounts of polypeptides with MM << 14 kDa (Figure 5, track 5).

### 3.9. Parameters of Enzyme Kinetics

The experiments were carried out using a chromogenic substrate that mimicked the Chn-sensitive region of bovine κ-CN [23]. Judging by the obtained Michaelis constants K_m_, the affinity of rChn-Alc-KL to the substrate was 2.2 and 3.7 times as low as those of rChn-Cam and rChn-Bos, respectively (Table 4).

Since rChn-Alc-KL reacted with the substrate more slowly than rChn-Bos and rChn-Cam, its k_cat_ and k_cat_/K_m_ were inferior to the corresponding parameters of their enzymatic efficiency. Under conditions of complete saturation with the substrate, the turnover number k_cat_ of rChn-Alc-KL was 2.6 and 1.5 times as low as that of rChn-Bos and rChn-Cam, respectively. Correspondingly, the efficiency k_cat_/K_m_, which is directly dependent on the MA of an enzyme, for rChn-Alc-KL was approximately 5.6 times as low as that for the commercial rChns. A comparison of the maximum rates V_max_ shows that rChn-Alc-KL provides the highest rate of hydrolysis of the chromogenic substrate.

Note that the relatively low values of k_cat_ and k_cat_/K_m_ obtained using a low-molecular-weight water-soluble synthetic substrate do not necessarily indicate a low efficiency of a rChn in the hydrolysis of actual κ-CN in casein micelles. For example, kcat and k_cat_/K_m_ of rChn-Cam obtained on a mimic synthetic substrate are 3.8 and 1.8 times lower than those of rChn-Bos. However, on cow’s milk, the coagulation activity of camel’s rChn is 70% higher than that of cow’s rChn [9].

### 3.10. Temperature Optimum

The dependence of the MA of rChn on the temperature of the substrate has a characteristic form: the MA gradually increases in the range of 25–50 °C and begins to decrease at T > 55 °C. As seen from Figure 6, all the studied rChns had the same T_opt_ and showed the maximum MA at a substrate temperature of 50–55 °C. With an increase in milk temperature above 55 °C, rChn-Bos was first to decrease its coagulation ability, rChn-Alc-KL was the last, and rChn-Cam behaved in between. All the rChns did not coagulate milk heated to 65 °C.

### 3.11. Dependence of Milk-Clotting Activity on pH of Milk

The MA optima of most known Chns lie in the acidic pH range; therefore, alkalization of milk leads to an increase in the duration of rennet clotting [45].

A response of the MA of the studied enzymes to an increase in substrate pH was different (Figure 7). With an increase in pH from 6.0 to 7.0, the coagulation activity of rChn-Alc-KL, rChn-Cam, and rChn-Bos decreased by factors of 4.3, 5.6, and 8.1, respectively. From the standpoint of cheese-making technology, this means that with the same total MA, at pH = 6.5 (pH of milk into which the coagulant was added), the consumption of rChn-Bos for coagulation of a unit volume of milk will be maximal, while the consumption of rChn-Alc-KL will be minimal. Such relatively low sensitivity of rChn-Alc-KL to milk pH in the range of 6.0–7.0 should be considered as a positive technological property.

Moose rChns obtained in *K. lactis* and *E. coli* [31] have a similar dependence of MA on pH.

### 3.12. Dependence of Milk-Clotting Activity on the Concentration of Calcium Chloride

The study of the sensitivity of the MA of a new rChn to the concentration of calcium chloride is important both for the general biochemical characterization of the enzyme and for understanding its technological prospects.

Most cheeses are made from pasteurized milk. Pasteurization of raw milk leads to the formation of almost insoluble calcium phosphate. As a result, the concentration of Ca ions in milk decreases, and this leads to a deceleration of rennet coagulation and a deterioration in the rheological characteristics of the clot. To avoid increasing the dose of an added MCE and improve the coagulation ability of pasteurized milk, CaCl_2_ is added to it at a concentration of 1–4 mM [21]. The addition of calcium chloride improves the coagulation properties of pasteurized milk for two reasons. First, Ca^2+^ partially shields the negative charge of the C-terminal regions of κ-CN, which form a “hair layer” on the surface of casein micelles. As a result, the Phe105-Met106 bond becomes more accessible to attack from the MCE. Second, Ca^2+^ is involved in the formation of ionic “bridges” between destabilized micelles, which accelerates their aggregation and the formation of a milk clot. The addition of CaCl_2_ is widely used in the cheese making to improve the technological properties of rennet-lean milk [12]. However, an increase in the concentration of CaCl_2_ causes an increase not only in the MA, but also in the total PA of the enzyme [50], which can negatively affect the quality of cheeses produced [48].

As seen from Figure 8, rChn-Alc-KL and rChn-Cam showed the same sensitivity to the concentration of calcium ions in milk. With an increase in the CaCl_2_ content from 0 to 5 mM, the coagulation capacity of the enzymes increased by 20–50%, while the capacity of rChn-Bos increased by 26–60%. Thus, compared with rChn-Bos, the coagulation activities of rChn-Alc-KL and rChn-Cam were less dependent on the concentration of calcium chloride. This means that with an increase in the amount of CaCl_2_ introduced into the cheese bath, rChn-Alc-KL and rChn-Cam will show a lower total PA than rChn-Bos will.

Comparing our results with the data of [31], we can conclude that the expression system does not affect the dependence of MA of rChn-Alc on the concentration of calcium chloride.

### 3.13. Influence of Divalent Metal Cations on Milk-Clotting Activity

Metal cations are among the factors that can significantly affect the activity of enzymes by changing the structure of their allosteric and active sites. In addition, metal cations can affect the rate of formation and decay of enzyme–substrate complexes. Nevertheless, the effect of divalent metal cations (Me^2+^) on the MA of rChn has been underexplored.

Indeed, prior to the publication in 2021 on *Camelus bactrianus* rChn [28], in numerous works devoted to rChn, this issue was raised only in one article [48] in 2012. Akishev et al. [28] showed that Ca^2+^; Fe^2+^; Ba^2+^, Mg^2+^, and Mn^2+^ increased the MA of *Camelus bactrianus* rChn expressed in *Komagataella* (*Pichia*) *pastoris* by 3.0, 3.4, 3.7, 1.9, and 5.2 times, respectively. On the contrary, Ni ions suppressed the activity of the enzyme, which led to a decrease in its MA by 5.0 times. A similar effect of Me^2+^, but to a much lesser extent, was also shown for rChn-Bos, also expressed in *K.* (*Pichia*) *pastoris* [61].

Table 5 shows the effect of divalent metal ions on the MA of rChn-Alc-KL and commercial rChns. As seen, the addition of Mg^2+^, Zn^2+^, Co^2+^, Ba^2+^, Fe^2+^, Mn^2+^, and Ca^2+^ to milk leads to an increase in the coagulation ability of rChns by 1.1–3.5 times. The greatest activating effect was shown by Mn^2+^. On the contrary, Ni ions suppressed the MA of the enzymes by 1.4–1.9 times. The effect of Cu ions turned out to be paradoxical; in their presence, the coagulation ability of rChn-Bos and rChn-Cam decreased by 26–32%, while the activity of rChn-Alc-KL on the contrary increased by 37%. Taking into account the high homology of the primary structure of studied rChns, the mechanism of such a different effect of Me^2+^ on their MA is not clear.

In general, our data on the effect of Me^2+^ on the MA of rChn are in good agreement with that in the literature [28,61]. In terms of practical application, these results can be used, for example, in formulating liquid preparations of recombinant milk coagulants for cheese making.

### 3.14. X-ray Diffraction Data

The X-ray diffraction study of single crystals of rChn-Alc-KL was carried out on the basis of 3600 diffractograms. It was found that this enzyme crystallizes in the C2 space group with the following unit cell parameters: a = 143.8 (5), b = 38.9 (2), c = 57.5 (2) Å, β = 90.64 (2)°. Unfortunately, it was not possible to solve the structure of the enzyme due to the lack of enough reflections; for 2510 protein atoms, there were 5002 unique reflections obtained, instead of the required minimum of 7531. To obtain diffraction data with a better quality, it will be necessary to optimize the crystallization conditions.

## 4. Conclusions

A moose rProChn producing strain—Alc-D *K. lactis*—was obtained using the hybrid autoinducible P_350_ promoter. Thus, the work demonstrated for the first time the possibility of using previously developed hybrid promoter [40] for the synthesis and secretion of moose prochymosin in *K. lactis*. The highest yield of the target enzyme was achieved when the producer was cultivated for 15 days and the culture medium contained 3% glucose, 1 g/L KH_2_PO_4_, and 1.6 mg/L calcium pantothenate. After activation of the zymogen, active rChn-Alc-KL preparations were obtained. The maximum coagulation activity of the enzyme in the c.f. achieved 86 AU/mL.

Recombinant Chn of moose was precipitated from the c.f. with ammonium sulfate and purified by hydrophobic and affinity chromatography. After partial purification, the relative MA (MA/A_280_) of the enzyme increased by a factor of 2450.

The biochemical properties of rChn-Alc-KL were compared with commercial rChn-Bos and rChn-Cam. In terms of specific MA, the enzyme of moose was superior to rChn-Bos, but inferior to rChn-Cam. The thermal inactivation threshold for rChn-Alc-KL was 55 °C; after heating at 60 °C, the enzyme was completely inactivated.

rChn-Alc-KL showed a higher total PA than the commercial enzymes. The MA/PA ratio for rChn-Alc-KL was 1.3 and 9.6 times as low as that for rChn-Bos and rChn-Cam.

In terms of affinity to the chromogenic mimic of the Chn-sensitive sequence of bovine κ-CN and in terms of enzymatic efficiency, rChn-Alc-KL was inferior than its commercial counterparts. Its Michaelis constant was 4.7 μM, which was 2.2–3.7-times higher than that of the reference enzymes. The parameters k_cat_ and k_cat_/K_m_ of rChn-Alc-KL were 98.7 s^−1^ and 21.1 µM^−1^ s^−1^ and were lower than those of the commercial rChns.

Compared to cow and camel rChns, an increase in milk pH from 6.0 to 7.0 suppressed the MA of rChn-Alc-KL to the least extent. In response to an increase in the concentration of CaCl_2_ from 0 to 5 mM, the coagulation ability of rChn-Alc-KL and rChn-Cam increased by 20–50%, and the ability of rChn-Bos increased by 26–60%.

rChn-Alc-KL, as well as the reference rChns, exhibited the maximum MA at a substrate temperature of 50–55 °C.

The addition of Mg^2+^, Zn^2+^, Co^2+^, Ba^2+^, Fe^2+^, Mn^2+^, and Ca^2+^ into milk led to an acceleration of its clotting by all three rChns by 1.1–3.5 times. Ni ions inhibited the activity of all the enzymes by 1.4–1.9 times. In the presence of Cu ions, the coagulation ability of rChn-Cam and rChn-Bos decreased by 26–32%, while the activity of rChn-Alc-KL on the contrary increased by 37%.

A comparison of the biochemical properties of moose rChn obtained in *E. coli* [31] and *K. lactis* (this study) showed that the replacement of the prokaryotic expression system with the eukaryotic one leads to an increase in the specific MA and coagulation specificity of the enzyme, but does not significantly affect its TS and the dependence of its MA on pH and calcium chloride concentration.

The purity and quality of the obtained rChn-Alc-KL made it possible to grow single crystals of this protein and use them for X-ray diffraction analysis. It was shown that the enzyme crystallizes in the C2 space group with the following unit cell parameters: a = 143.8 (5), b = 38.9 (2), c = 57.5 (2) Å, β = 90.64 (2)°. To solve the structure of rChn-Alc-KL, it is necessary to optimize the crystallization conditions and increase the array of diffraction reflections.

## Figures and Tables

**Figure 1 foods-12-03772-f001:**
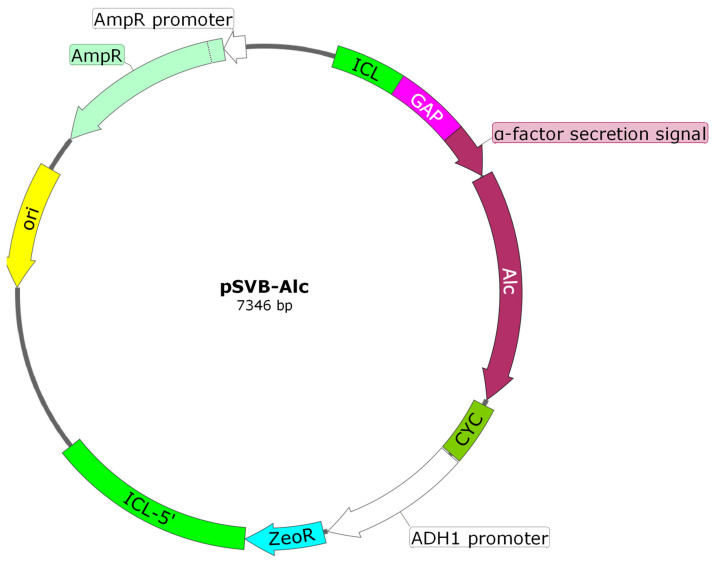
Genetic map of the pSVB-Alc plasmid vector: ICL—regulatory sequence of isocitrate lyase promoter; GAP—core sequence of glyceraldehyde-3-phosphate dehydrogenase promoter; α-factor secretion signal—α-MCE *S. cerevisiae* protein secretion signal; Alc—moose prochymosin sequence; CYC—CYC1 *S. cerevisiae* transcription terminator sequence; ADH1—alcohol dehydrogenase gene promoter; ZeoR—zeocin resistance gene; ICL-5′—5′ isocitrate lyase promoter gene sequence; ori—the replication origin; AmpR—ampicillin resistance gene.

**Figure 2 foods-12-03772-f002:**
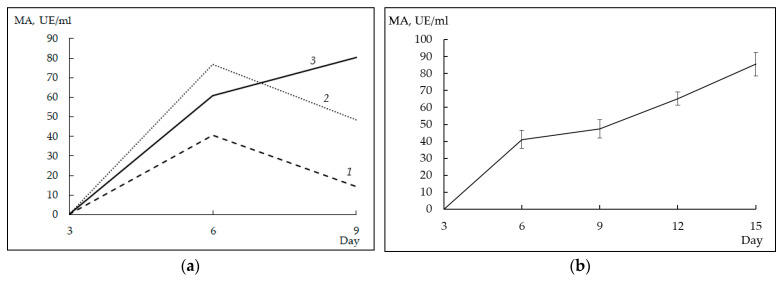
Productivity of producer strain (Alc-D *K. lactis)* in terms of moose rProChn under various cultivation conditions: (**a**) dependence of milk-clotting activity on glucose concentration: 1—series 1 (1% glucose); 2—series 2 (2% glucose); 3—series 3 (3% glucose); (**b**) typical dependence of MA on the duration of cultivation at the initial glucose concentration of 3%.

**Figure 3 foods-12-03772-f003:**
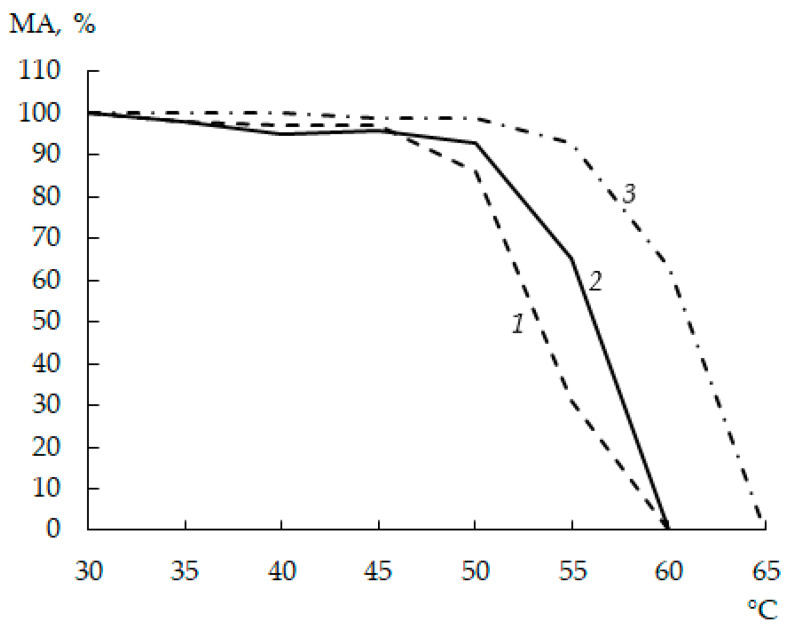
Milk-clotting activity of recombinant chymosins (rChns) from different animal sources (1—rChn-Bos; 2—rChn-Alc-KL; 3—rChn-Cam) at different temperatures (30–65 °C).

**Figure 4 foods-12-03772-f004:**
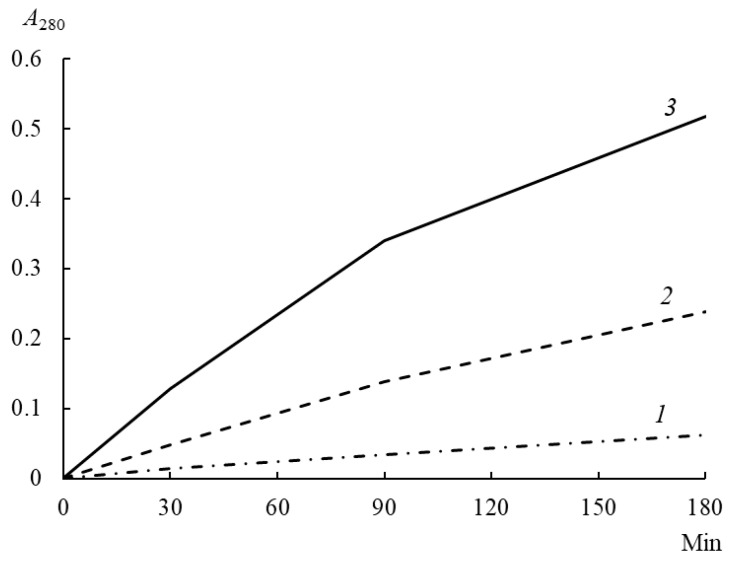
Total proteolytic activity (A_280_) of recombinant chymosins (rChn) (1—rChn-Cam; 2—rChn-Bos; 3—rChn-Alc-KL) after different incubation periods (0–180 min) of enzyme–substrate mixtures.

**Figure 5 foods-12-03772-f005:**
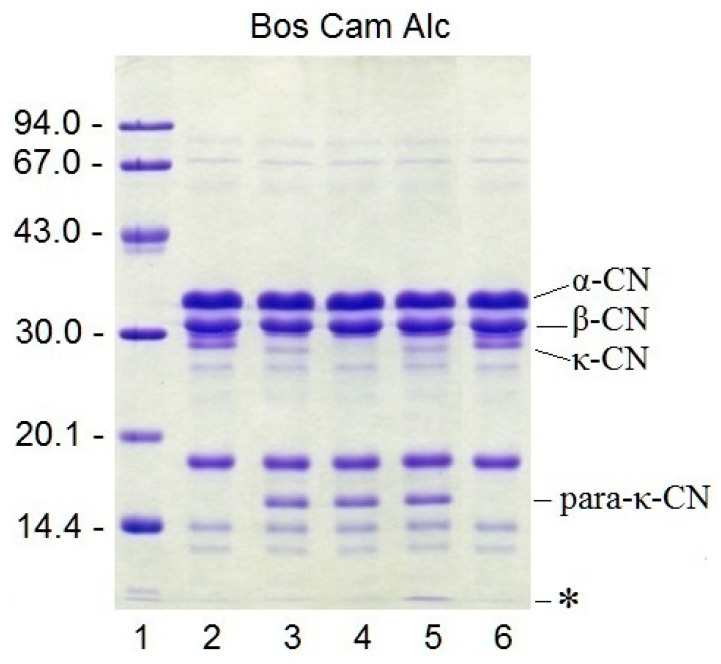
Proteolytic specificity of recombinant chymosins: 1—molecular weight markers; 2—control 1 (substrate without heating); 3—milk + rChn-Bos; 4—milk + rChn-Cam; 5—milk + rChn-Alc-KL; 6—control 2 (substrate heated at 35 °C for 60 min). On the left, molecular weights are indicated in kDa. On the right, the bands of α-, β-, κ-CN, and para-κ-CN (MM ≈ 16 kDa, tracks 3–5) are indicated. An asterisk (*) shows polypeptide components with MM << 14 kDa (track 5).

**Figure 6 foods-12-03772-f006:**
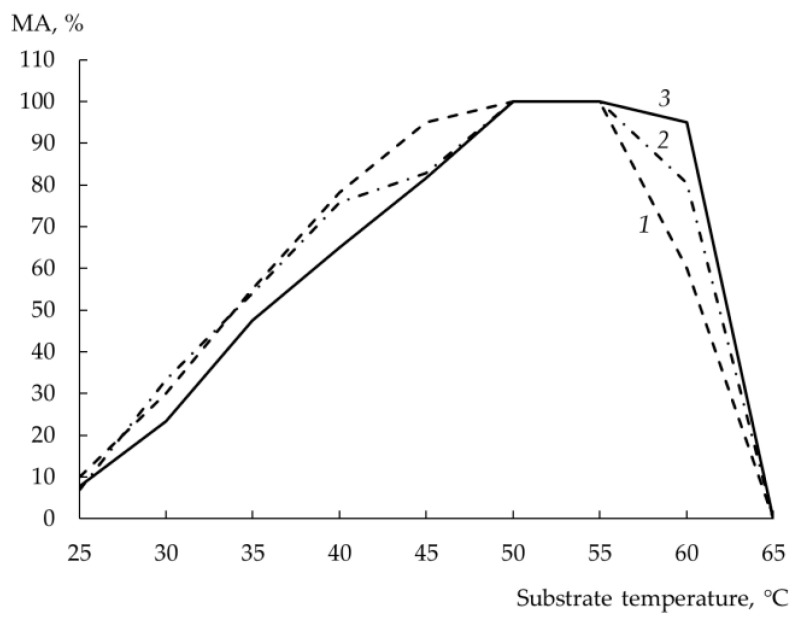
Milk-clotting activity (MA, %) of recombinant chymosins (rChns) (1—rChn-Bos; 2—rChn-Cam; 3—rChn-Alc-KL) at various substrate temperatures (25–65 °C).

**Figure 7 foods-12-03772-f007:**
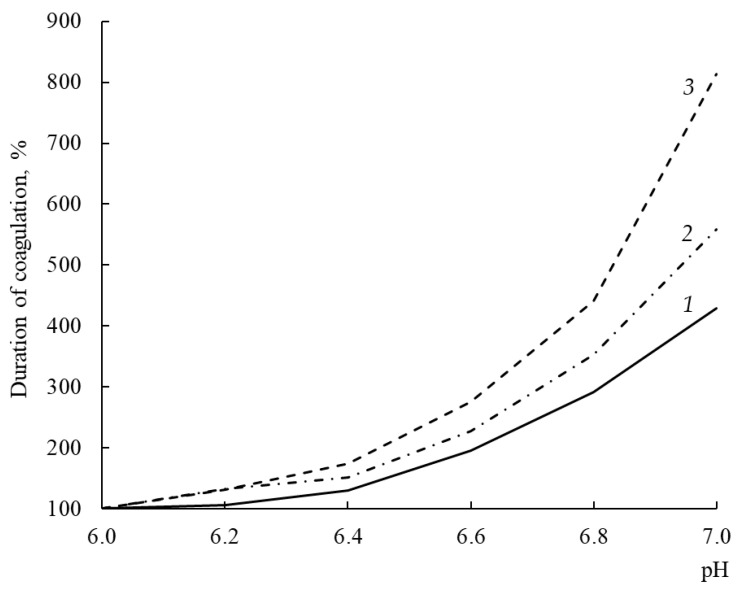
Milk-clotting activity (in terms of duration of coagulation, %) of recombinant chymosins (rChns) (1—rChn-Alc-KL; 2—rChn-Cam; 3—rChn-Bos) at various pH (6.0–7.0) of the milk substrate.

**Figure 8 foods-12-03772-f008:**
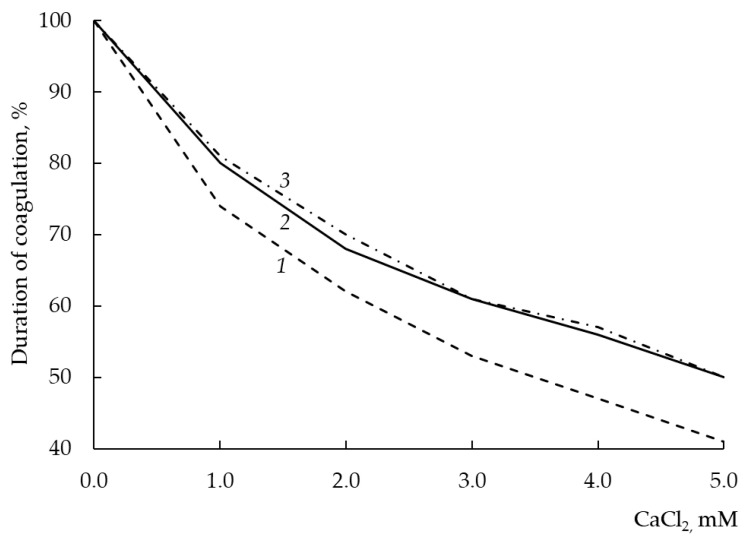
Milk-clotting activity (in terms of duration of coagulation, %) of recombinant chymosins (rChns) (1—rChn-Bos; 2—rChn-Cam; 3—rChn-Alc-KL) at various concentrations of calcium chloride (0.0–5.0 mM) in milk.

**Table 1 foods-12-03772-t001:** Physical and biochemical parameters of moose rChn preparations before and after partial purification. Abbreviations decoding: rChn-Alc—recombinant chymosin of *A. alces*, obtained in the *K. lactis* production system; MA—milk-clotting activity; A_280_—absorption at a wavelength of 280 nm.

Preparation	Volume (mL)	MA (AU/mL)	Total MA (AU)	A_280_	Relative MA (MA/A_280_)
Culture liquid with rChn-Alc-KL	4800	86	412,800	9.798	8.8
Partially purified rChn-Alc-KL	15	14,469	217,035	0.669	21,627.8

**Table 2 foods-12-03772-t002:** General and specific milk-clotting activity of recombinant chymosin preparations. Abbreviations decoding: rChn-Alc—recombinant chymosin of *A. alces*, obtained in the *K. lactis* production system; rChn-Bos—recombinant chymosin of *B. taurus*; rChn-Cam—recombinant chymosin of *C. dromedarius*; MA—milk-clotting activity.

Preparation	Total MA (AU/mL)	Protein Concentration (mg/mL)	Specific MA (AU/mg)	Specific MA (%)
rChn-Alc-KL	1293 ± 16	0.082 ± 0.001	15,768	106
rChn-Bos	1293 ± 16	0.087 ± 0.001	14,862	100
rChn-Cam	1324 ± 0	0.077 ± 0.001	17,195	116

**Table 3 foods-12-03772-t003:** Specific milk-clotting activity, total proteolytic activity, and coagulation specificity (MA/PA) of recombinant chymosins (rChn). Abbreviations decoding: rChn-Alc—recombinant chymosin of *A. alces*, obtained in the *K. lactis* production system; rChn-Bos—recombinant chymosin of *B. taurus;* rChn-Cam—recombinant chymosin of *C. dromedarius*; MA—milk-clotting activity; PA—proteolytic activity.

rChn	Specific MA (%)	PA (A_280_)	PA (%)	MA/PA
rChn-Alc-KL	106	0.332 ± 0.001	138	0.8
rChn-Bos	100	0.241 ± 0.001	100	1.0
rChn-Cam	116	0.037 ± 0.000	15	7.7

**Table 4 foods-12-03772-t004:** Michalis–Menten kinetic parameters of recombinant chymosins (rChns). Abbreviations decoding: rChn-Alc—recombinant chymosin of *A. alces*, obtained in the *K. lactis* production system; rChn-Bos—recombinant chymosin of *B. taurus*; rChn-Cam—recombinant chymosin of *C. dromedarius*; K_m_—Michaelis constant; V_max_—maximum rate of the enzymatic reaction; k_cat_—turnover number of the enzyme; k_cat_/K_m_—catalytic efficiency.

rChn	K_m_, μM	V_max_, nM/s	k_cat_ (s^−1^)	k_cat_/K_m_, (μM^−1^ s^−1^)
rChn-Alc-KL	4.69 ± 0.27	616.85 ± 28.44	98.69 ± 4.55	21.11 ± 0.78
rChn-Bos	2.12 ± 0.09	315.31 ± 3.48	252.24 ± 2.79	119.58 ± 4.74
rChn-Cam	1.27 ± 0.16	190.30 ± 29.21	152.24 ± 23.37	118.81 ± 6.74

**Table 5 foods-12-03772-t005:** Effect of divalent metal cations (Me^2+^) on the milk-clotting activity of recombinant chymosins.

Substrate (±Me^2+^)	MCA (%)		
	rChn-Alc-KL	rChn-Bos	rChn-Cam
Control	100.0	100.0	100.0
Cu^2+^	136.6 ± 2.1	67.9 ± 1.3	73.5 ± 1.6
Mg^2+^	259.6 ± 3.7	204.1 ± 7.2	162.5 ± 1.8
Ni^2+^	73.3 ± 0.6	71.3 ± 1.4	53.0 ± 0.7
Zn^2+^	161.1 ± 5.0	109.0 ± 2.3	111.5 ± 4.4
Co^2+^	256.2 ± 7.0	201.8 ± 9.0	145.0 ± 2.5
Ba^2+^	319.3 ± 9.7	296.6 ± 10.8	246.4 ± 0.0
Fe^2+^	308.1 ± 5.3	296.0 ± 5.7	224.3 ± 5.9
Mn^2+^	386.8 ± 8.4	308.1 ± 10.5	321.6 ± 6.8
Ca^2+^	298.4 ± 10.1	296.6 ± 10.8	217.7 ± 6.2

## Data Availability

The datasets generated for this study are available on request to the corresponding author.

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
