# Peer review of "Biochemical Properties of a Promising Milk-Clotting Enzyme, Moose (Alces alces) Recombinant Chymosin"

_foods, 2023, doi:10.3390/foods12203772_

Round 1

Reviewer 1 Report

The manuscript aimed to invest the biochemical properties of a promising milk-clotting enzyme. The work has some degree of novelty, application value and scientific rigour, which can bring absolute inspiration. Thus, this paper needs minor revision before publication. Some important issues highlighted in the following comments need to be addressed entirely.

1)      The author could supplement the gene sequence alignment information of the ProChn-Alc and other chymosin genes. There should be a comparative analysis and discussion from a genetic perspective.

2)      It is recommended to supplement the molecular weight, purification table, and SDS-PAGE of the recombinant enzyme.

3)      The X-ray diffraction results do not seem to have much meaning under this measurement result. Thus, homologous modeling and comparative analysis can be considered.

Minor editing of English language required, For example, some sentences are too long to be easily understood. There are also some grammar and tense issues in this manuscript.

Author Response

The team of authors would like to thank the reviewer for his appreciation of our work and important comments.

Comments and Suggestions for Authors:

The manuscript aimed to invest the biochemical properties of a promising milk-clotting enzyme. The work has some degree of novelty, application value and scientific rigour, which can bring absolute inspiration. Thus, this paper needs minor revision before publication. Some important issues highlighted in the following comments need to be addressed entirely.

1)      The author could supplement the gene sequence alignment information of the ProChn-Alc and other chymosin genes. There should be a comparative analysis and discussion from a genetic perspective.

In our opinion, comparative genetic analysis of chymosin genes has low information value, since it does not allow a connection between the properties of the enzyme and the individual nucleotide sequence of the gene. At the same time, comparative analysis of amino acid sequences is valuable. We conducted such an analysis in our previous article (Balabova, D. V., Rudometov, A. P., Belenkaya, S. V., Belov, A. N., Koval, A. D., Bondar, A. A., ... & Shcherbakov, D. N. (2022). Biochemical and technological properties of moose (Alces alces) recombinant chymosin. Vavilov Journal of Genetics and Breeding, 26(3), 240). We decided not to include this analysis in this article to avoid repetition.

2)      It is recommended to supplement the molecular weight, purification table, and SDS-PAGE of the recombinant enzyme.

We do perform SDS-PAGE analysis of chymosins during enzyme purification processes.

   1       2       3        4        5        6        7

70 kDa

35 kDa

15 kDa

1 - Thermo Scientific™ PageRuler™ Plus Prestained Protein Ladder, 10 to 250 kDa;

2 - 6 – chromatographic fractions;

7 – purified rChn-Alc-KL.

However, since the purpose of this work was rather to study the biochemical and technological properties of this enzyme, we decided not to include this figure in the text of the article.

3)      The X-ray diffraction results do not seem to have much meaning under this measurement result. Thus, homologous modeling and comparative analysis can be considered.

We completely agree with the remark. We actually previously performed homology modeling in our previous paper (Balabova, D. V., Rudometov, A. P., Belenkaya, S. V., Belov, A. N., Koval, A. D., Bondar, A. A., ... & Shcherbakov, D. N. (2022). Biochemical and technological properties of moose (Alces alces) recombinant chymosin. Vavilov Journal of Genetics and Breeding, 26(3), 240). However, we included a section on X-ray diffraction in this article because in this way we wanted to show that the protein under study was truly homogeneous, which even allowed for crystallization and the results obtained were determined by the presence of the target protein and not impurities. In the future, we plan to publish these results in a separate article.

Reviewer 2 Report

This manuscript is about milk-clotting enzyme from moose expressed in yeast, and I have some questions:

- I do not understand why it is important the chymosyn gene from moose. In text, the gene from cow and camel were better, were not?

- I miss, for comparison, the analysis of milk-clotting enzyme from moose without expressing in yeast, just to know if the properties were altered

Author Response

The team of authors would like to thank the reviewer for his appreciation of our work and important comments.

Comments and Suggestions for Authors:

This manuscript is about milk-clotting enzyme from moose expressed in yeast, and I have some questions:

- I do not understand why it is important the chymosin gene from moose. In text, the gene from cow and camel were better, were not?

From the point of view of cheese-making, rChns of cow and camel are high-quality, but not ideal technological enzymes - each of them has both advantages and disadvantages. For example, the rChn of a cow is inferior to the rChn of a camel in the ratio of milk-clotting and proteolytic activity (MA/PA), but has a technologically more advantageous range of thermal stability (Belenkaya, S.V. Basic Biochemical Properties of Recombinant Chymosins (Review) / S.V. Belenkaya, D.V. Balabova, A.N. Belov, A.D. Koval, D.N. Shcherbakov, V.V. Elchaninov // Applied Biochemistry and Microbiology.- 2020.- V. 56.- № 4.- P. 363-372). The goal of our group is to search for the perfect milk-clotting enzyme among various groups of Mammals. We assumed that moose rChn obtained in the eukaryotic expression system (K.lactis) could compete with cow and camel rChns. Unfortunately, this did not happen. But this allows us to exclude the moose rChn from further searches.

- I miss, for comparison, the analysis of milk-clotting enzyme from moose without expressing in yeast, just to know if the properties were altered

Good question. Unfortunately, we didn't have the opportunity to obtain natural chymosin (Chn) from the gastric mucosa of a baby moose, which feeds exclusively on milk. Nowadays, it is sometimes easier to obtain a recombinant enzyme than its natural analogue. The results of published studies comparing natural cow Chn and its genetically engineered version indicate that the main biochemical properties of the enzymes differ slightly. In addition, the industrial production of natural moose Chn is impossible, due to the lack of a raw material base (no one harvests the mucous membranes of the stomachs of newborn moose calves). Based on this, we set the task to study the biochemical and technological properties of A. alces rChn obtained in the K.lactis expression system.

Reviewer 3 Report

The objective and significance of the study

The goal of the current study was to prepare recombinant chymosin (rChn) from  Kluyveromyces lactis and study its biochemical specifications. The study opted this system for its attractive properties;  the Crabtree-negative carbohydrate metabolism, the ease of scaling, the GRAS status, the possibility of cultivation using simple culture media, and the availability of highly efficient promoters. The obtained product was  a convenient tool for obtaining heterologous recombinant proteins both on a laboratory and industrial scale.  So the goal may be technologically significant.

Since Kluyveromyces lactis is a well known source for the production of chymosin, the study should clarify the novelty. It should explain the difference of chymosin produced in the previous research and technologies and its production with the new technique adopted in the article.  It should specifically clarify the potential advantages of the new system compared to the old ones. This should be clear in the introduction he discussion.

Materials and Methods

This section needs to be reorganized to be compatible with the next section (Result and discussion).

The zymogen (rProChn-Alc.) was mentioned 4 times (pages 3, 4 &5) without explaining its name or identity. Please, explain it at its first appearance since it is not a common construct. This term should be distinct from the term of the K. lactis production system (rChn-Alc-KL) to avoid any confusion for the reader.

In page 4 section Zymogen activation please mention what is the product of activation. rProChn-Alc was activated according to the procedure described in [23].

In page 4 after the section (Partial purification of moose rChn) a new section should be introduced titled (Biochemical characterization of rChn). All the following analytical methods should be sequentially regrouped under this section.

Page 14 change (rsing) into (using)

Illustrations

The quality of Figure 1 should be improved to be more readable and conceivable.

Modify the title of Figure 2 into (Figure 2. Productivity of the producer strain (Alc-D K. lactis) in terms…….).

Tables 1-4: Explain all the abbreviations so that the tables can be independently self-explained. 

Figure 3, modify the title into (Figure 3. Milk-clotting activity of recombinant chymosins (rChn) from different animal sources (1 – rChn-Bos; 2 – rChn-Alc-KL; 3 – rChn-Cam) at different temperatures (30- 65 °C).

Figure 4, modify the title into (Figure 4. Total proteolytic activity (A280) of recombinant chymosins (rChn)  (1 – rChn-Cam; 2, rChn-Bos; 3 – rChn-Alc-KL) after different  incubation periods of enzyme-substrate mixtures.)

Modify the other figures (6, 7, 8) accordingly.

Minor linguistic modifications

Page 2,

In the sentence (From 2001 to the present, the following rChns has been obtained and characterized: sheep (Ovis aries) [13]); change (has) to (have).

Change (which demonstrated a number of biochemical properties) into (which demonstrated some biochemical properties )

Change (the rChn zymogen synthesized in E. coli is accumulated) into (the rChn zymogen synthesized in E. coli accumulates)

Change (hinders the obtaining of an active enzyme.) into (hinders obtaining an active enzyme..)

Page 5

Under section 3.1

Change (upon depletion of glucose in the culture medium.) into (upon glucose depletion in the culture medium.)

Change (medium, which contained acetamide) into (medium, containing acetamide)

Under section 3.2

Modify (The initial glucose content in the culture medium has a significant impact on the density of the yeast culture and its productivity in terms of the target protein.) into (The initial glucose content in the culture medium significantly impacts the yeast culture's density and its productivity in terms of the target protein.)

Moderate English editing is preferred

Author Response

The team of authors would like to thank the reviewer for his appreciation of our work and important comments.

The objective and significance of the study

The goal of the current study was to prepare recombinant chymosin (rChn) from  Kluyveromyces lactis and study its biochemical specifications. The study opted this system for its attractive properties;  the Crabtree-negative carbohydrate metabolism, the ease of scaling, the GRAS status, the possibility of cultivation using simple culture media, and the availability of highly efficient promoters. The obtained product was  a convenient tool for obtaining heterologous recombinant proteins both on a laboratory and industrial scale.  So the goal may be technologically significant.

Since Kluyveromyces lactis is a well known source for the production of chymosin, the study should clarify the novelty. It should explain the difference of chymosin produced in the previous research and technologies and its production with the new technique adopted in the article.  It should specifically clarify the potential advantages of the new system compared to the old ones. This should be clear in the introduction he discussion.

We completely agree with the remark, but our article is devoted rather to studying the properties of a new enzyme, moose chymosin. As a producer, we tried to choose an already known chymosin production system in order to be able to compare with existing results. However, we used a new hybrid promoter and the relevant information is included in the conclusion section.

Thus, the work demonstrated for the first time the possibility of using the previously developed hybrid promoter [40] for the synthesis and secretion of moose prochymosin in K. lactis.

Materials and Methods

This section needs to be reorganized to be compatible with the next section (Result and discussion).

The zymogen (rProChn-Alc.) was mentioned 4 times (pages 3, 4 &5) without explaining its name or identity. Please, explain it at its first appearance since it is not a common construct. This term should be distinct from the term of the K. lactis production system (rChn-Alc-KL) to avoid any confusion for the reader.

Thanks for the comment. But the abbreviation rProChn-Alc is deciphered at its first mention on page 3: "As a result, a producer of recombinant moose prochymosin (rProChn-Alc) …."

In page 4 section Zymogen activation please mention what is the product of activation. rProChn-Alc was activated according to the procedure described in [23].

Thank You for valuable comment. On page 4, the phrase has been added: As a result of activation, K.lactis c. f. containing rChn-Alc-KL was obtained).

In page 4 after the section (Partial purification of moose rChn) a new section should be introduced titled (Biochemical characterization of rChn). All the following analytical methods should be sequentially regrouped under this section.

Thank You so much for valuable comment. A new section - Biochemical characterization of rChn – has been added.  All the following analytical methods sequentially regrouped under this section. 

Page 14 change (rsing) into (using)

Thank You very much! Done.

Illustrations

The quality of Figure 1 should be improved to be more readable and conceivable.

Thank You very much! Done.

Modify the title of Figure 2 into (Figure 2. Productivity of the producer strain (Alc-D K. lactis) in terms…….).

Thank You very much! Done.

Tables 1-4: Explain all the abbreviations so that the tables can be independently self-explained. 

Thank You for valuable comment. Abbreviations have been added to the headings of all tables:

Table 1. …  Abbreviations decoding:  rChn-Alc-KL - recombinant chymosin of A. alces, obtained in the K. lactis production system; МА – Milk-clotting activity; A280 - absorption at a wavelength of 280 nm.

Table 2. … Abbreviations decoding:  rChn-Alc - recombinant chymosin of A. alces, obtained in the K. lactis production system; rChn-Bos - recombinant chymosin of B. taurus; rChn-Cam - recombinant chymosin of C. dromedarius; МА – milk-clotting activity.

Table 3. … Abbreviations decoding:  rChn-Alc - recombinant chymosin of A. alces, obtained in the K. lactis production system; rChn-Bos - recombinant chymosin of B. taurus; rChn-Cam - recombinant chymosin of C. dromedarius; МА – milk-clotting activity; PA - proteolytic activity.

Table 4. … Abbreviations decoding:  rChn-Alc - recombinant chymosin of A. alces, obtained in the K. lactis production system; rChn-Bos - recombinant chymosin of B. taurus; rChn-Cam - recombinant chymosin of C. dromedarius; Km – Michaelis constant; Vmax - maximum rate of the enzymatic reaction; kcat - turnover number of the enzyme; kcat/Km - catalytic efficiency.

Figure 3, modify the title into (Figure 3. Milk-clotting activity of recombinant chymosins (rChn) from different animal sources (1 – rChn-Bos; 2 – rChn-Alc-KL; 3 – rChn-Cam) at different temperatures (30-65 °C).

Thank You very much! Done.

Figure 4, modify the title into (Figure 4. Total proteolytic activity (A280) of recombinant chymosins (rChn)  (1 – rChn-Cam; 2, rChn-Bos; 3 – rChn-Alc-KL) after different  incubation periods (0-180 min) of enzyme-substrate mixtures.)

Thank You very much! Done.

Modify the other figures (6, 7, 8) accordingly.

Thank You very much! Done.

Figure 6. Milk-clotting activity (MA, %) of recombinant chymosins (rChn) (1 – rChn-Bos; 2 – rChn-Cam; 3 – rChn-Alc-KL) at various substrate temperatures (25-65 oC).

Figure 7. Milk-clotting activity (in terms of Duration of coagulation, %) of recombinant chymosins (rChn) (1 – rChn-Alc-KL; 2 – rChn-Cam; 3 – rChn-Bos) at various pH (6.0-7.0) of the milk substrate.

Figure 8. Milk-clotting activity (in terms of Duration of coagulation, %) of recombinant chymosins (rChn) (1 – rChn-Bos; 2 – rChn-Cam; 3 – rChn-Alc-KL) at various concentrations of calcium chloride (0.0-5.0 mM) in milk.

Minor linguistic modifications

Page 2,

In the sentence (From 2001 to the present, the following rChns has been obtained and characterized: sheep (Ovis aries) [13]); change (has) to (have).

Thank You very much! Done.

Change (which demonstrated a number of biochemical properties) into (which demonstrated some biochemical properties )

Thank You very much! Done.

Change (the rChn zymogen synthesized in E. coli is accumulated) into (the rChn zymogen synthesized in E. coli accumulates)

Thank You very much! Done.

Change (hinders the obtaining of an active enzyme.) into (hinders obtaining an active enzyme..)

Thank You very much! Done.

Page 5

Under section 3.1

Change (upon depletion of glucose in the culture medium.) into (upon glucose depletion in the culture medium.)

Thank You very much! Done.

Change (medium, which contained acetamide) into (medium, containing acetamide)

Thank You very much! Done.

Under section 3.2

Modify (The initial glucose content in the culture medium has a significant impact on the density of the yeast culture and its productivity in terms of the target protein.) into (The initial glucose content in the culture medium significantly impacts the yeast culture's density and its productivity in terms of the target protein.)

Thank You very much! Done.

Round 2

Reviewer 1 Report

The author revised the manuscript and improved its quality of writing and presentation. Based on this situation, I have no other questions or suggestions for now.

Quality of English Language is ok.

Author Response

Thank you very much for your valuable comments!

Reviewer 3 Report

The authors responded to the comment on novelty by inserting the following  part  in the conclusion (Thus, the work demonstrated for the first time the possibility of using the previously developed hybrid promoter [40] for the synthesis and secretion of moose prochymosin in K. lactis). However, this should be also indicated in the introduction to prepare the reader for understanding the work.

Please change Escherichia coli into italic in page 2 (twice) and page 3 (once).  

All other required changes have been conducted appropriately.

Author Response

Thank you very much for your comments. They are very valuable to our work.

The authors responded to the comment on novelty by inserting the following  part  in the conclusion (Thus, the work demonstrated for the first time the possibility of using the previously developed hybrid promoter [40] for the synthesis and secretion of moose prochymosin in K. lactis). However, this should be also indicated in the introduction to prepare the reader for understanding the work.

We have added a sentence: "To obtain the enzyme, in our work we designed an original expression cassette that included a previously developed hybrid promoter [40]. And for the first time, demonstrated the possibility of using this promoter for the synthesis and secretion of moose prochymosin in K. lactis"  in the introduction section.

Please change Escherichia coli into italic in page 2 (twice) and page 3 (once).  

We have made the appropriate changes.